# The Evaluation and Conservation of Plant Genetic Resources Collected in Lithuania



Denise F. Dostatny [1,*], Aleksandra Korzeniewska [2], Grzegorz Bartoszewski [2], Ryszard Rawski [3], Karolina Kaźmińska [2] and Bronislovas Gelvonauskis [4]

1 National Centre for Plant Genetic Resources, Plant Breeding and Acclimatization Institute, National Research Institute, Radzików, 05-870 Błonie, Poland
2 Department of Plant Genetics, Breeding and Biotechnology, Institute of Biology, Warsaw University of Life Sciences, Nowoursynowska, 159, 02-776 Warsaw, Poland; aleksandra_korzeniewska@sggw.edu.pl (A.K.); grzegorz_bartoszewski@sggw.edu.pl (G.B.); karolina_kazminska@sggw.edu.pl (K.K.)
3 Polish Academy of Sciences Botanical Garden, Center for Biological Diversity Conservation in Powsin, ul. Prawdziwka, 2, 02-973 Warsaw, Poland; pomologia@obpan.pl
4 Plant Gene Bank, Stoties 2, Akademija, 58343 Kedainiai, Lithuania; b.gelvonauskis@agb.lt
* Correspondence: d.dostatny@ihar.edu.pl

**Abstract:** The present work compiles the results of three-year expeditions organized between 2011 and 2013 aimed at plant crop collection in the area of Lithuania, an Eastern European country. Accessions of fruit trees, vegetables, cereals, forage, industrial, fibrous, medical, and spice crops as well as accompanying segetal plants were collected in 5 ethnographic regions of Lithuania. In total, 1010 samples of seeds, bulbs, and plant grafts were obtained. The majority of the collected samples belonged to the Cucurbitaceae, Fabaceae, Solanaceae, Alliaceae, and Rosaceae families. The accessions were described and deposited in the long-term storage in Lithuanian and Polish Gene Banks. Almost all collected cucumbers, pumpkins, squashes, and oat plants were morphologically characterized in field experiments. *Cucurbita pepo* accessions showed high morphological diversity, while limited diversity of cucumber accessions was observed. Oat plants were characterized by high morphological diversity and resistance to diseases present in some of the investigated accessions. Further characterization of collected apple trees and other accessions is underway. Obtaining germplasm in the area of Lithuania fulfills a gap in current plant crop collections. Collected material could be valuable for pre-breeding evaluation and further breeding programs as well as the study of genetic diversity.

**Keywords:** agrobiodiversity; collection missions; gene bank; genetic erosion; evaluation



## 1. Introduction

The idea of collecting, keeping, and using plant genetic resources dates back to the beginning of the 20th century. Modern agriculture is based on genetically uniform crops which have led to genetic erosion of cultivated plants [1]. Therefore, in the 21st century, we still need to organize collection missions in order to preserve the loss of the plant genetic resource diversity caused by changes in agriculture methods and the transformation of the landscape. Until the diversity of crops and their wild relatives is mostly collected and preserved, expeditions should continue.

In Lithuania, the first crop collections were established at the end of the 19th century. In 1886, Hrebnicki established a pomological collection in the north-eastern region of Lithuania [2]. The collection of agricultural crops and plant genetic resources started when the Dotnuva plant breeding station was founded in 1922 [3]. At that time, collections of various agricultural crops, e.g., rye, barley, oat, potato, clover, forage grasses, and others, were established and breeding programs of these crops were started [3]. Vegetable crop cultivar testing and breeding programs were initiated in 1923. Fruit cultivar testing and breeding at the experimental station for horticulture were started in the late 1940s [4].

In Lithuania, the National Plant Genetic Resources (PGR) program was founded in 1993 with the Lithuanian Research Center for Agriculture and Forestry as its coordination center. The Baltic-Nordic Plant Genetic Resources project was initiated by the Nordic Gene Bank (NGB) in 1994. The main objective of these projects was to develop the national PGR conservation network in Lithuania. Eight institutions are involved in the activity of collection, investigation, and conservation of plant genetic resources. The Government of the Republic of Lithuania decided to establish the Gene Bank on the 1 January 2004. Its main functions include coordination activities for the collection, research, conservation, and use of the national plant genetic resources in Lithuania and preservation of the genetic material in long-term storage [5].

The area for the collection of the plant genetic resources in this study is connected with the historical association of Lithuania and Poland [6]. This area was to a certain extent inhabited by the Poles; therefore, similar cultivation methods and practices can be observed in some regions of Lithuania. In those times, an extensive exchange of seed material and information concerning running a homestead occurred. Thus, the joined search for common gene resources in these two countries.

During expeditions, besides passport data, the collectors recorded valuable knowledge shared by farmers about traits that they valued in the plants, and the ways they cultivated, harvested, and processed them. This information is a treasure trove of data that can also help us understand the consequences of climate changes and changes in agricultural practices better [7].

The purpose of field expeditions to Lithuania was to collect seeds, bulbs, and plant grafts for further propagation and evaluation, and to deposit the collected accessions in long-term storage and in the collection of living plants both in the National Center for Plant Genetic Resources at Plant Breeding and Acclimatization Institute—the National Research Institute (NCPGR-PBAI-NRI) in Radzików, Poland and in the Lithuanian Gene Bank.

## 2. Study Area

Lithuania is situated on the east coast of the Baltic Sea. Distance from the Baltic Sea coast to the eastern border is 373 km and from the north to south, 276 km. The area of Lithuania is 65.3 -km$^2$ and its agricultural area is 29.5 km$^2$ [8]. The lowlands make up approximately 75% of the area. Because of the geographic location, the climate is transitional, maritime and continental. The average monthly temperature in winter (December, January, February) is $-2.8\ °C$, in spring (March, April, May) 6.5 $°C$, in summer (June, July, August) 16.8 $°C$ and in autumn (September, October, November) 7.1 $°C$. The Baltic Sea's influence on the climate is observed only in the coastal region. The temperature in autumn and winter is 2–3 $°C$ higher in the west than in the east while in spring and summer it is relatively higher in the east. The soil temperature in winter is 0.5 $°C$ lower than the air temperature, and in summer it is 3–6 $°C$ higher whereas the highest soil surface temperature is observed on the coast. Furthermore, the vegetation period in the west lasts for 145–160 days and is shorter in the eastern parts of Lithuania. Annual precipitation varies from 560 to 910 mm with an average of 695 mm. The smallest amount of precipitation falls in April and February (36–38 mm) and the highest in July and August (77 mm). The highest yearly precipitation amount falls on the western and south-western Žemaičiai Upland whereas the smallest falls on the north-eastern parts of this region. Climate changes observed in the last decade (2001–2010) caused an increase in the average air temperature, mostly in the spring months and in July. Furthermore, climate anomalies, including heavy rainfalls, are becoming more frequent [9].

According to the World Bank, agricultural land (% of land area) in Lithuania was reported at 47.16% in 2016 [10]. About 2.5 million ha constitute land under agricultural crops (corn, forage grasses, and pastures). Large (apple, pear, sour cherry) and small (strawberry, black and red currants, raspberry, high bush blueberry) fruits trees are grown commercially in more than 8.5 thousand ha.

Soil types vary from region to region in Lithuania [11]. Low-fertility acid soddy podzolic sandy loams as well as sands and drained podzolic gleys occupy about 63% of the area. Fertile soddy calcareous, predominantly loams, and drained soddy gleys cover 26% of the area. Four regions in terms of soil types are identified. In the central part of Lithuania, the most productive soils are located; the western part has wetter acid soils; the east of the country has sandy hills and woodlands; in northern areas, there is a region of calcareous type soils. The soil type in the regions where the material was collected is varied. More details about the distribution of soil types are provided by Jukneviciute and Laurinavicius [11].

Lithuania is divided into 5 ethnographic regions: Aukštaitija, Dzūkija, Mažoji Lietuva, Suvalkija, and Žemaitija (Figure 1).

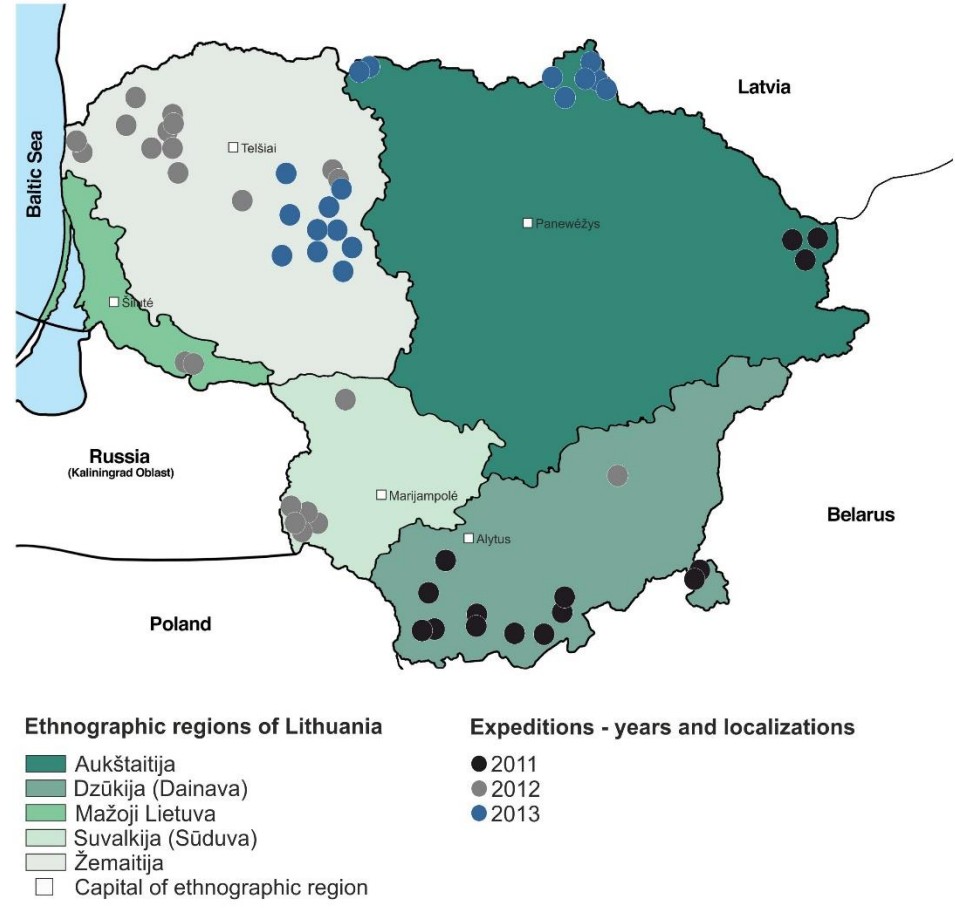

**Figure 1.** Ethnographic regions of Lithuania.

The collected accessions were classified according to these regions because they better illustrated the agricultural and horticultural practices in this country. Below is a brief description of individual ethnographic regions (based on [12–15]).

*2.1. Aukštaitija*

It is the biggest and the oldest region of Lithuania including the areas of central and north-eastern Lithuania. The region lies in the Vidurio Lietuvos Lowland (which creates a widespread marshy area cut with numerous river valleys and through which the biggest river of the country, the Neman, flows) and in the Baltijos Lake District in the drainage basin of the upper and middle Neris, on the Niesvies, the Šventoji, and the Mūša. People live in 'linear settlements' here, which means that the houses stand next to each other along the streets, and the rest of the farmstead is situated behind the houses. This might be the

result of a very diversified landscape, the multitude of hills, lakes, and forests, owing to which each village has become a separate world of its own.

### 2.2. Dzūkija

The south-eastern part of Lithuania, called Dzūkija or Dainava, has a different local dialect and a distinctive culture. Most of this region belongs to the Alytus District, which is characterized by the lowest population density and the highest percentage of forest cover. In this part of Lithuania, there are many areas protected under the EU Nature 2000. Because of the higher quality of soils, agriculture plays an important role there. The life of many people in Dzūkija is still connected with the forest. Dzūkija was not subject to strong civilization influences and social changes as was the case in other parts of Lithuania. Owing to this, one can find here villages with traditional wooden architecture located near the mosaic of plots, pastures, and meadows lying by the rivers.

### 2.3. Mažoji Lietuva

Mažoji Lietuva, a seaside region with unique nature, is considered to be the cradle of Lithuanian writing. At present, it includes the Klaipėda District (Kursiu Nerija sandbar), the Šilutė and Klaipėda area as well as the southern part of the Taurage District. The landscape here is undulating, full of hills, lakes, and forests. The area is famous for the production of cottage cheeses made of produce obtained from own farmsteads. Various spices are added to obtain more diversified flavors.

### 2.4. Suvalkija (Sūduva)

A region in the southern and south-western part of Lithuania, beyond the Neman River. This is the ethnic land of the 'Sūduva people', for whom agriculture has been the main occupation since the beginning of the first millennium. With a very high level of agriculture, it is one of the most important agricultural regions of Lithuania. Sūduva is a region of plains and fertile soils, with big, independent, family-owned farmsteads. Among the wide fields, one can see scattered homesteads surrounded by trees. The farmsteads are distant from each other, and they are self-sufficient. In farmsteads, we can find houses, gardens, ponds, orchards, animals, and fields as well as greenhouses with vegetables. There is a tradition of baking bread on calamus leaves and producing cheese in this region.

### 2.5. Žemaitija

This is a region in the north-western part of Lithuania. In geographical terms, it constitutes the Žemaitija Lake District with moraines which are built of clays, sands, and marls. The soils are of medium fertility. There are many marshes and lakes formed by the glacier. The hilly and wavy surface is covered with cultivable fields, meadows, pastures, and spruce forests.

## 3. Materials and Methods

### 3.1. Procedure for the Collection of Plant Genetic Material

Three independent expeditions were organized in 2011, 2012, and 2013. They were organized taking the seed-ripening time into account and were conducted in September and October. Seeds, bulbs, and grafts were collected and interviews with farmers were conducted. Precise information was also taken down, concerning the place of collection, i.e., the name of the region, village, details of the farmer, geographic coordinates, altitude. If an accession was collected from a wild state, its site was described [16].

Seeds were obtained mainly from farmers' warehouses or collected in the field. At the time of the collection, the accessions were given expedition acronyms. For the expedition performed in 2011, the given acronym was LITLIT11 (and in sequence the number of the collected accession). For the expedition performed in 2012—LITZAP12 and for the expedition performed in 2013—LITCEN13. In order to keep the article concise and coherent,

these acronyms have been abbreviated to the last three letters and the year: LIT11, ZAP12, and CEN13.

During the collection of material, passport data for each accession were completed and documented. In the case of wild-type harvest, the site was described [17]. The collection of genetic material was made in accordance with international and national regulations and the Standard Material Transfer Agreement (SMTA) was assigned [18]. A part of the collected material has been forwarded to the Lithuanian Gene Bank and the second part is kept at National Center for Plant Genetic Resources at the Plant Breeding and Acclimatization Institute—National Research Institute (NCPGR-PBAI-NRI), Radzików, Poland.

A part of the cereal accessions was collected in the form of blends. During the cleaning of these materials, they were divided into separate accessions, according to the belonging species. When the study area was explored, most of the plants accompanying cultivation, segetal plants, had already been damaged in the fields after the harvest. Because of this, in order to identify this plant group, seeds in the collected cereal crop accessions, both from blends and from individual species, were marked. Separated seeds were identified according to Kulpa [19].

*3.2. Processing Collected Plant Genetic Material*

After returning from the expeditions, all information was entered in the database of the National Center for Plant Genetic Resources (PBAI-NRI), while the accessions, together with their passport data, were sent for propagation and evaluation to the curator of a given plant group. After that, seeds of separate plant groups were returned to the Gene Bank and stored at a low temperature in the long-term storage located at NCPGR-PBAI-NIR [20]. The procedures were conducted according to the "Genebank Standards for Plant Genetic Resources for Food and Agriculture" [21].

3.2.1. Initial Evaluation of Cucurbit Plants

Collected accessions of cucurbit crops were planted in 2014 at 'Wolica' Experimental Station of the Department of Plant Genetics, Breeding and Biotechnology of the Warsaw University of Life Sciences (DPGBB-WULS), Warsaw, Poland. Initially, morphological and performance characteristics of 53 collected accessions of cucurbits were described; the accessions belonged to: cucumber *Cucumis sativus* L. (30) and pumpkin *Cucurbita pepo* L. (23). To this end, 15 plants from each accession were planted in the field. The accessions of cucumber were evaluated with regard to the following features: sex expression, mature fruit traits, i.e., length and width, skin, and spine color. Sex expression and fruit spines color were described according to [22]. As a reference, the 'Trakai' variety was used. Seeds of the 'Trakai' variety were obtained from the NCPGR-PBAI-NRI (accession number PL172603). For *C. pepo*, fruit weight, yield per plant, skin color, and seed type were evaluated. The varietal type was described according to [23] and the other traits according to the directions of the European Cooperative Programme for Plant Genetic Resources [24]. As a reference, the Polish cultivar 'Danka Polka' pumpkin was used. Seeds of 'Danka Polka' were obtained from W. Legutko seed company (W. Legutko, Jutrosin, Poland).

3.2.2. Initial Evaluation of Oat Plants

Collected accessions of *Avena sativa* L., e.g., nine in 2011 and nineteen in 2012 were planted in the plots of the Plant Breeding and Acclimatization Institute (PBAI-NRI) in 2012–2013. In total, 600 plants/accessions were sown manually in 2.5 m$^2$ plots. Two times during the growing season of plant lodging, emerging diseases were noted. The plant height of 10 individuals from each plot was measured and averaged. The thousand grains weight (TGW) and yield in grams were calculated. Two Polish varieties—'Celer' (from Małopolska Hodowla Roślin, Kraków, Poland) and 'Krezus' (from Hodowla Roślin Strzelce, Strzelce, Poland)—served as the references (in each year, 7 plots with Polish reference varieties were planted). Lodging of plants in the plots was observed (scale 0–9, where 9 means no lodging) and emerging diseases (septoriosis, powdery mildew, and

crown rust). The descriptors were used according to International Board for Plant Genetic Resources (IBPGR) [25]. The resistance scale was used instead of the susceptibility scale (where 9 is a healthy non-lodging plant, and 1 refers to a less resistant, lodging plant) according to the guidelines used in Poland (COBORU) [26].

### 3.2.3. Initial Evaluation of Apple Trees

The species vegetatively propagated were placed in the Gene Bank plantations as living plant collections in the Botanical Garden in Powsin (Powsin—Poland). For fruit trees, mainly apple trees, documentation was prepared during the expedition, i.e., fruit photographs and descriptions, after which multiplication material was collected (i.e., grafts), and also thoroughly described and photographed. Fruit descriptions were prepared according to the International Union for the Protection of New Varieties of Plants (UPOV) descriptors used for variety characteristics of newly-bred crop plant species varieties. From the several dozen features included in the descriptors, the appearance of trees, leaves, flowers, and fruits among others, the most important 24 features to describe fruits were chosen.

### 4. Results

During three expeditions organized in the years 2011, 2012, and 2013 in 5 ethnographic regions of Lithuania, seeds, bulbs, and grafts of 1010 accessions belonging to 5 plant groups were collected and described (Table 1). The highest number of accessions was collected in the Žemaitija region; it constituted 39.4% of all accessions. A big number of accessions was collected in the Aukstaitija Region: 24.5% and Dzūkija Region: 21.5%. In other regions, fewer accessions were collected: in Suvalkija 10.0% and in Mažoji Lietuva 4.7%. Among the accessions collected, the biggest group was constituted by local populations or local varieties of vegetable plants—632 accessions and it constituted as much as 62.5% of all collected accessions. A lot of fruit trees and bushes varieties (13.7%, 139 accessions) and cereal plants (13.6%, 125 accessions) were collected. Also, 43 accessions of medical and spice plants were collected (4.7%); this share was bigger than in the case of forage (34 accessions, 3.3%) and industrial crops (24 accessions, 2.2%) (Table 1). Together with cereals seeds, several segetal species were collected (38 species of plants, which were not included in Table 1).

**Table 1.** Accessions collected during three expeditions to Lithuania in 2011, 2012, and 2013.

| Species | Number of Accessions Collected in Each Ethnographic Region | | | | | Total |
|---|---|---|---|---|---|---|
| | Aukštaitija | Dzūkija | Mažoji Lietuva | Suvalkija | Žemaitija | |
| **Fruit trees and bushes** | | | | | | |
| *Malus × domestica* Borkh. | 44 | 43 | 1 | | 39 | 127 |
| *Malus sylvestris* (L.) Mill. | | 6 | 1 | | 3 | 10 |
| *Rosa* sp. | | | | | 2 | 2 |
| **Vegetable crops** | | | | | | |
| **Annual** | | | | | | |
| *Anethum graveolens* L. | 2 | | 1 | 1 | 10 | 14 |
| *Capsicum annuum* L. | 11 | | 3 | 3 | 2 | 19 |
| *Cucurbita maxima* Duchesne | 4 | 21 | 3 | 1 | 10 | 39 |
| *Cucurbita pepo* L. | 14 | 22 | 3 | 1 | 15 | 55 |
| *Cucumis sativus* L. | 9 | 17 | 1 | 4 | 16 | 47 |
| *Citrullus lanatus* (Thumb.) Matsum & Nakai | | | 1 | | | 1 |

**Table 1.** *Cont.*

| Species | Number of Accessions Collected in Each Ethnographic Region | | | | | Total |
|---|---|---|---|---|---|---|
| | Aukštaitija | Dzūkija | Mažoji Lietuva | Suvalkija | Žemaitija | |
| *Glycine max* (L.) Merr. | | | | 1 | 1 | 2 |
| *Lactuca sativa* L. | 2 | | | 5 | 1 | 8 |
| *Phaseolus vulgaris* L. | 49 | 21 | 2 | 22 | 35 | 129 |
| *Phaseolus coccineus* L. | 5 | | 4 | 1 | 8 | 18 |
| *Solanum lycopersicum* L. | 39 | 5 | 9 | 6 | 39 | 98 |
| *Vicia faba* L. var. *major* Harz. | 11 | 1 | | | 18 | 30 |
| **Biennial and perennial** | | | | | | |
| *Allium ampeloprasum* ssp. *porrum* J. Goy | | | | | 1 | 1 |
| *Allium cepa* L. var. *cepa* Helm. | 7 | 14 | 1 | 3 | 26 | 51 |
| *Allium cepa* L. var. *aggregatum* G. Don. | 3 | | | | 6 | 9 |
| *Allium × proliferum* (Moench) Schard. | | | 1 | | 4 | 5 |
| *Allium sativum* L. | 21 | 10 | 4 | 12 | 34 | 81 |
| *Allium fistulosum* L. | 1 | | | | | 1 |
| *Allium schoenoprasum* L. | 1 | | | 1 | | 2 |
| *Brassica napus* L. var. *napobrassica* L. Peterm. | | | | | 1 | 1 |
| *Petroselineum sativum* L. ssp. *crispum*. | | | | | 4 | 4 |
| *Brassica oleracea* L. var. *capitata* L. f. alba | 1 | | | | 1 | 2 |
| *Beta vulgaris* L. | 2 | 1 | 1 | 1 | 4 | 9 |
| *Daucus carota* L. | 1 | | | | | 1 |
| *Heliantus tuberosus* L. | | | | | 3 | 3 |
| *Pastinaca sativa* L. | | 1 | | | | 1 |
| *Rheum rhaponticum* L. | | | | | 1 | 1 |
| **Cereals *, forage, industrial, fibrous and other crops** | | | | | | |
| *Pisum sativum* L. | 4 | | | 2 | 12 | 18 |
| *Hordeum vulgare* L. | 4 | | | 6 | 21 | 31 |
| *Fagopyrum esculentum* Moench. | | 11 | | | 2 | 13 |
| *Linum usitatissimum* L. | | 1 | | | 3 | 4 |
| *Trifolium pratense* L. | | 1 | | 1 | 2 | 4 |
| *Vicia sativa* L. | | 1 | | 1 | | 2 |
| *Canabis sativa* L. | | 6 | 2 | | 1 | 9 |
| *Melilotus sp.* L. | | 1 | | | 2 | 3 |
| *Triticum aestivum* L. | 5 | 7 | 1 | 6 | 12 | 31 |
| *Triticum spelta* L. | | | | | 1 | 1 |
| *× Triticosecale* Wittm. | | 2 | | 5 | 7 | 14 |
| *Lupinus albus* L., *L. luteus* L., *L. angustifolius* L. | | 2 | 1 | | 1 | 4 |

**Table 1.** *Cont.*

| Species | Number of Accessions Collected in Each Ethnographic Region | | | | | Total |
|---|---|---|---|---|---|---|
| | Aukštaitija | Dzūkija | Mažoji Lietuva | Suvalkija | Žemaitija | |
| *Avena sativa* L. | 3 | 7 | | 6 | 16 | 32 |
| *Secale cereale* L. | | 8 | 1 | 2 | 4 | 15 |
| *Nicotiana rustica* L. | 1 | | 2 | 1 | 2 | 6 |
| *Nicotiana tabacum* L. | | | | | 1 | 1 |
| *Raphanus sativus* L. | | | | 1 | | 1 |
| *Medicago sativa* L. | | | | 1 | 1 | 2 |
| *Brassica nigra* (L.) Koch | | | | 1 | | 1 |
| *Betula vulgaris* ssp. *vulgaris* convar. *crossa* Alef. var. *rapa* | | | 1 | | | 1 |
| *Baptisia australis* (L.) R. Br. | | | | | 1 | 1 |
| *Zea mays* L. | | | | | 1 | 1 |
| *Helianthus annuus* L. | | 1 | | | | 1 |
| **Medical and spice crops** | 3 | 7 | 3 | 6 | 24 | 43 |
| **Subtotal/total** | 247 | 217 | 47 | 101 | 398 | 1010 |

\* Some of cereals were collected as a mixture.

### 4.1. Apple Trees

Despite the climatic conditions which are not favorable for growing fruit crops, a relatively big pool of old fruit trees was observed (at the age of about 80–100 years), mainly apple trees and—to a smaller extent—pear trees. Varieties that were cataloged and collected in the form of grafts are largely those which had been grown in Lithuania or had come from neighboring countries with similar climates. The health of many trees was poor, and the situation is made even worse by the fact that the absolute majority of them grow in abandoned homesteads or in former, now neglected, manor orchards.

Grafts of 132 apple tree varieties from 47 habitats were collected. Thirty-one (31) varieties of known names were cataloged, 14 of which were added to the collection of the Botanical Garden in Powsin. The majority of varieties, due to the lack of fruits or the presence of fruits that were not clearly characteristic, remained unmarked, i.e., the appearance of the fruits did not allow to determine the variety in an explicit way. Fruits of the unmarked plants were thoroughly described and photographed, which will facilitate marking in the future.

During the collection mission carried out in 2011, the orchard implemented by Hrebnicki was visited. It is located in the Rous Estate near Dukštas in the Aukstaitija Region. Apart from known apple tree varieties occurring almost throughout Europe, there were some very rare ones known only in Lithuania, bred or described by Hrebnicki, among others 'Ananas Berżenicki', 'A la Napoleon', 'Długotrwałe', 'Gruszowka Hrebnickiego', 'Pepina Jana', 'Szlachcic', which were collected.

Another interesting site visited during the collection mission in 2011 was an orchard started in 1936 in the Pogarenda habitat (Varėnos region) by Ivanauskas (1882–1970), a biologist and co-founder of the Botanical Garden in Kaunas. The orchard was located deep in a forest, just next to the border with Belarus. At present, the place is abandoned. Earlier, there was a forester's lodge there, in which a few families lived—12 varieties of apple tree accessions were collected in that orchard.

Apple trees collected and cataloged in the territory of Lithuania can be divided into three groups. The first is constituted by varieties bred or found in the area of Lithuania. The following varieties belong to this group: 'Cukrówka Litewska', 'Malinówka Bierżenicka', 'Pepina Jana', 'Śmietankowe', 'Synap Biełoruskij', 'Reneta Litewska', 'Talve Nauding',

'Meelis' and others. The second group is made of varieties coming also from the countries bordering the Baltic Sea, but they often occur in other European countries; these include, among others: 'Ananas Berżenicki', 'Strumiłłówka' (Sierinka), different varieties of 'Antonówka', 'Oliwka Żółta' (Papierówka), 'Suislepskie', 'Glogierówka' (Pepinka Litewska), 'Truskawkowe Nietschnera', 'Charłamowskie' and 'Czarnoguz'. The third group consists of varieties coming from Poland, Germany, France, England, or the Netherlands. This group includes 'Grafsztynek Inflancki'—a Dutch variety frequent in the whole area of Lithuania, an equally popular French variety—'Kronselska', a Polish variety—'Kosztela', a German variety—'Reneta Landsberska', or French varieties—'Kalwila Biała Zimowa' and 'Reneta Szara Francuska'.

In Table 2, 14 varieties are listed, divided according to their occurrence into individual ethnographic regions. These are varieties chosen from 132 varieties marked during the cataloging in the area of Lithuania which have proper names. Most of them were bred or found in the area of the countries bordering the Baltic Sea and they are best adapted to the climatic and soil conditions in the area. Additionally, the varieties were grouped by regions in which they occurred. Some apple fruits and trees representing collected accessions of *Malus × domestica* Borkh. are shown in Figure 2.

**Table 2.** Varieties of apple trees according to the ethnographic region of the collection.

| Ethnographic Region | Varieties |
| --- | --- |
| Aukštaitija (Vilnius Region) | Barchatnoje, Malinówka Bierżenicka, Paniemuńskie Białe, Reneta Litewska, Uspiech |
| Dzūkija (Alytaus Region) | Cukrówka Litewska |
| Žemaitija (Šiaulių Region) | Birutes Pepinas, Lofen, Meelis, Pepina Jana, Synap Bieloruskij, Śmietankowe, Talvenauding |
| Žemaitija (Raseinių Region) | Pepina Czernienko |

*4.2. Vegetables*

A total of 632 accessions of vegetable crops were collected during the three expeditions in the following regions: Žemaitija—240 accessions, Aukstaitija—183 accessions, and Dzūkija—113 accessions. A lower number of accessions was collected in the Suvalkija Region (62), and the fewest in the Mažoji Lietuva Region (34). The majority of the collected accessions were annual (73%) and the others were biennial and perennial (27%) vegetable species (Table 1). The 460 annual vegetable accessions belonged to five families: Fabaceae (28.4%), Cucurbitaceae (22.5%), Solanaceae (18.5%), Apiaceae (2.2%), and Asteraceae (1.3%). The other 172 accessions of biennial and perennial vegetable crops belonged to the families Alliaceae (23.6%), Chenopodiaceae (1.43%) Apiaceae (0.95%), Brassicaceae (0.48%), Asteraceae (0.47%), and Polygonaceae (0.16%). Varieties and ecotypes from the families Fabaceae (179), Cucurbitaceae (142) and Solanaceae (117) dominated among the collected accessions of annual vegetable plants. Less numerous local populations of annual vegetable plants were observed in families Apiaceae (14) and Asteraceae (8 accessions). Fourteen (14) local populations of dill (*Anethum graveolens* L.) were collected, of which 10 accessions came from the Žemaitija Region, as well as 8 local populations of lettuce (*Lactuca sativa* L.) coming from the Suvalkija Region (5 accessions) (Table 1). Examples of collected vegetables are shown in Figure 3.

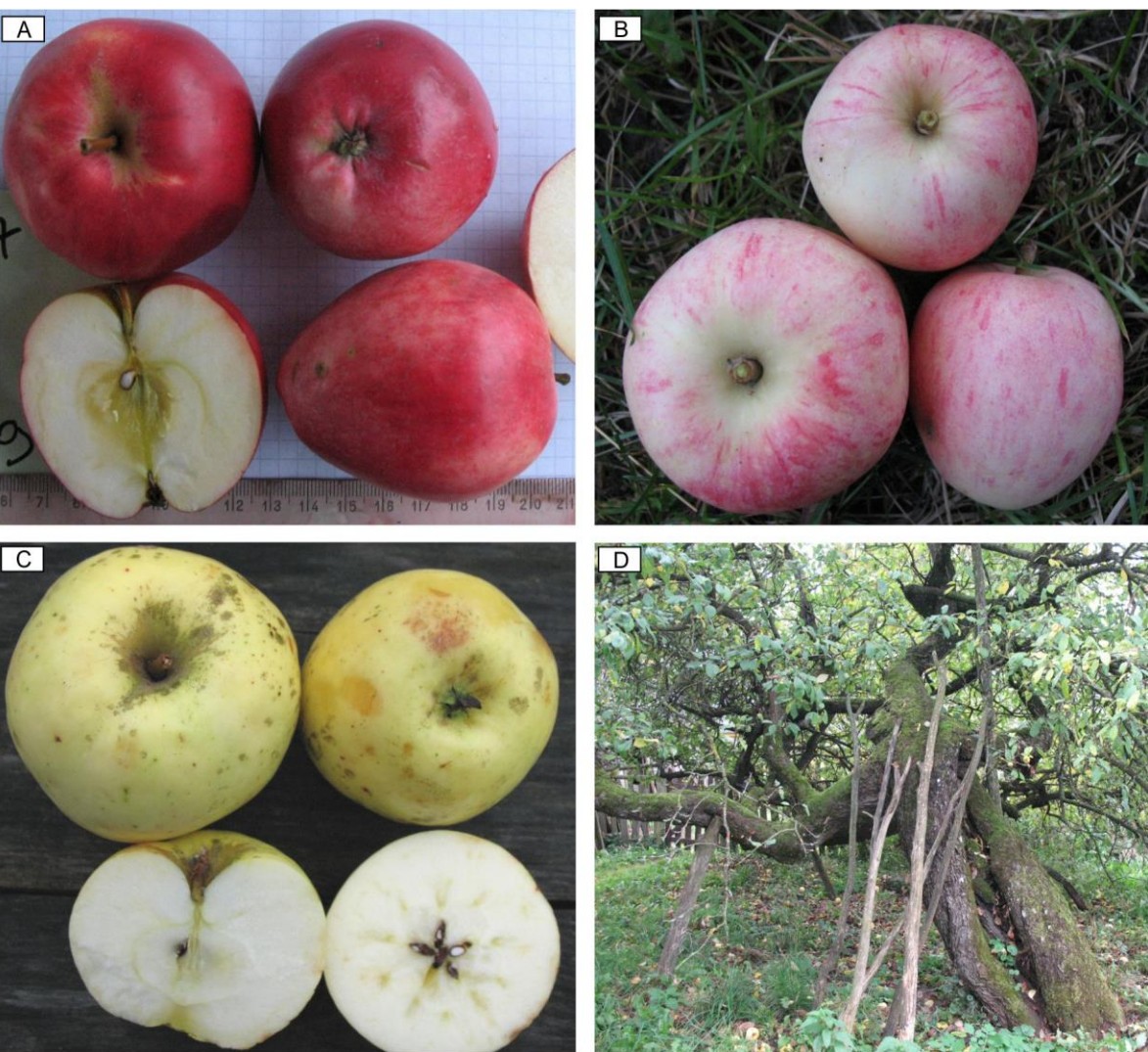

**Figure 2.** Apple fruits and a tree representing collected accessions of *Malus* × *domestica* Borkh. Fruits of 'Pepina Jana' (**A**), 'Śmietankowe' (**B**), 'Cukrówka Litewska', (**C**) and a tree of 'Cukrówka Litewska' (**D**).

#### 4.2.1. Cucurbit Vegetables

Local varieties and populations from the Cucurbitaceae family (142 altogether) constituted an equally numerous group among the collected accessions belonging to annual vegetable plants (Table 1). The group included: *Cucurbita pepo* L. (55), *Cucumis sativus* L. (47), *Cucurbita maxima* Duchesne (39), and *Citrullus lanatus* (Thumb.) Matsum & Nakai (1). The biggest number of accessions belonging to this family came from the Dzūkija Region: *C. pepo* (22), *C. maxima* (21), and *C. sativus* (17). In this region, cucurbits were grown in the majority in home gardens.

To characterize the biological diversity of *C. pepo*, 23 accessions were selected and examined in a field experiment conducted in 2014. The accessions came from various regions of Lithuania: Dzūkija (14), Aukstatija (5), Žemaitija (3), and Suvalkija (1). It was found that the examined accessions belonged to four different varietal groups of *C. pepo*. The majority of evaluated accessions were populations of pumpkins, vegetable marrow, zucchini, or a mixture of pumpkin with vegetable marrow and zucchini. A single accession of the scallop was identified (LIT11 265). High diversity of evaluated morphological features was observed within individual accessions collected. The most frequently observed fruit skin color was dark green or green with yellow or orange spots or stripes, plain or with a different number of small bumps. There were also fruits with

cream-colored and orange skin. The evaluated accessions varied in terms of average fruit weight, which ranged from 1.0 kg (LIT11 265) to 4.4 kg (LIT11 014). High differences of fruit weight within the accessions were also noticed, e.g., from 1.8 to 6.4 kg (LIT11 013) or from 0.6 to 4.3 kg (LIT11 159). The fruit yield per plant ranged for evaluated accessions from 4.5 kg (LIT11 266) to 15.3 kg (LIT11 013) with very high differences within the accession, e.g., from 2.0 to 21.1 kg (LIT11 100). The seeds of *C. pepo* accessions were of various sizes—small, medium, large, and very large, hulled or segregating for hulled and hull-less (Table 3).

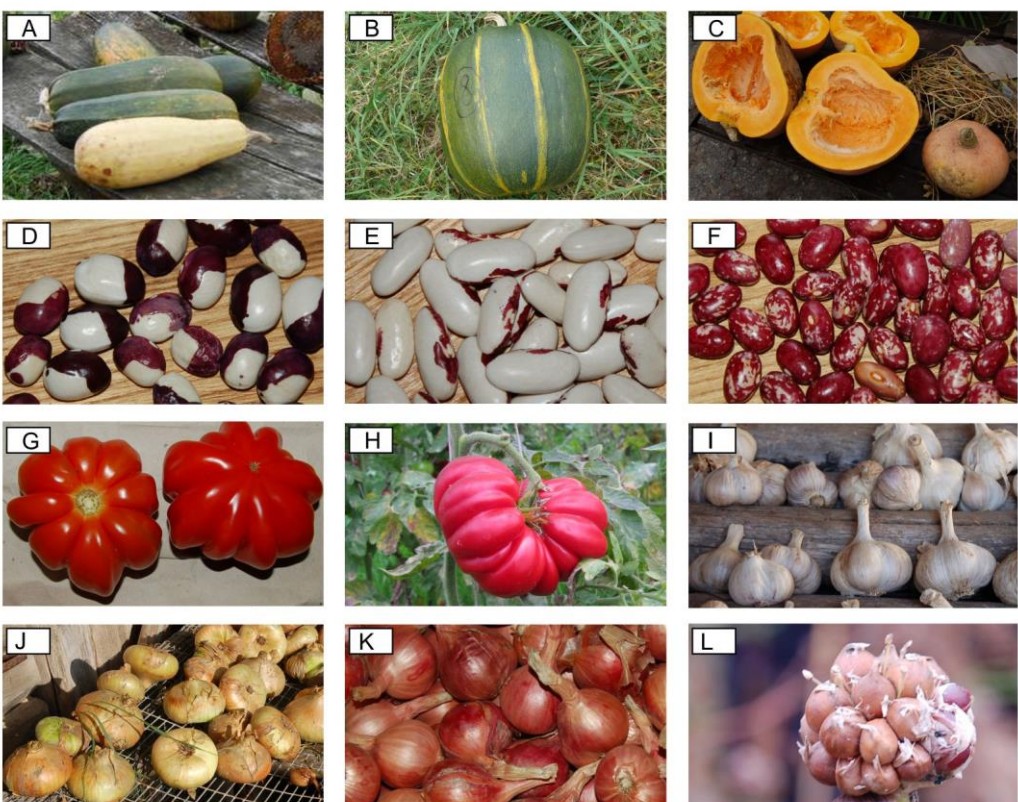

**Figure 3.** Examples of vegetables collected in Lithuania during the expedition in 2012. Vegetables belonging to the Cucurbitaceae (**A–C**), Fabaceae (**D–F**), Solanaceae (**G,H**) and Alliaceae families (**I–L**).

The second species of the Cucurbitaceae family, which is very frequently found in Lithuania, especially in the Dzūkija (17 accessions) and Žemaitija (16 accessions) regions, was cucumber (*Cucumis sativus* L.). Mature fruit traits and sex expression of 30 accessions were characterized in the field experiment (Table 4). Plants of a single accession were gynoecious (LIT11 034), whereas plants of four accessions were segregated for monoecious and gynoecious (LIT11 035, LIT11 128, LIT11 147, and LIT11 161), while the plants of 25 remaining accessions were monoecious. Limited variability of fruit features was observed within the examined accessions. Mature fruits showed cream or brown skin color and for many accessions, segregation for this trait within accession was observed. The color of fruit spines was white or dark (=brown or black) and also segregation for this trait within accession was observed. The length of mature fruit ranged from 17.0 to 25.6 cm with an average of 18.7 cm and 17.3 cm for the 'Trakai' reference variety. The fruits of collected accessions, except two of them (LIT11 010 and LIT11 257), were similar in length and also similar to fruits of the 'Trakai' variety. Fruit diameter ranged from 6.4 to 8.6 cm with an average of 7.6 cm and 8.3 cm for reference. The majority of the tested accessions were similar to the 'Trakai' variety of cucumber which was developed in Lithuania in the region of Trakai for open-field cultivation.

**Table 3.** Varietal type, fruit, and seed traits of several collected *Cucurbita pepo* L. accessions.

| Accession Number | Varietal Type | Fruit Skin Color | Fruit Weight (kg) | Total Fruit Yield Per Plant (kg) | Seed Types |
|---|---|---|---|---|---|
| LIT11 013 | pumpkin, vegetable marrow | dark green, orange | 2.9 ± 1.4 | 15.3 ± 7.0 | hulled |
| LIT11 014 | pumpkin | dark green | 4.4 ± 0.7 | 7.6 ± 1.8 | hulled |
| LIT11 036 | pumpkin, vegetable marrow | green, cream | 3.1 ± 1.2 | 9.8 ± 3.9 | hulled |
| LIT11 037 | pumpkin, vegetable marrow, zucchini | dark green, orange | 3.7 ± 2.0 | 9.3 ± 6.2 | hulled |
| LIT11 066 | zucchini | dark green | 2.5 ± 1.1 | 6.9 ± 3.0 | hulled |
| LIT11 074 | pumpkin | dark green | 4.1 ± 0.8 | 13.5 ± 4.7 | hulled |
| LIT11 098 | pumpkin | orange | 2.8 ± 1.1 | 11.0 ± 4.5 | hulled |
| LIT11 100 | vegetable marrow | dark green, green | 2.8 ± 0.5 | 13.7 ± 5.4 | hulled |
| LIT11 101 | pumpkin | dark green, green, orange | 3.5 ± 0.7 | 11.1 ± 5.5 | hulled, hull-less |
| LIT11 129 | pumpkin | green, cream | 1.5 ± 0.2 | 8.7 ± 2.8 | hulled |
| LIT11 157 | zucchini | dark green | 2.6 ± 0.6 | 7.6 ± 2.2 | hulled |
| LIT11 158 | vegetable marrow | cream | 2.7 ± 0.8 | 6.1 ± 3.2 | hulled |
| LIT11 159 | pumpkin | green, dark green | 2.4 ± 0.7 | 6.7 ± 3.0 | hulled |
| LIT11 208 | pumpkin | dark green | 2.1 ± 0.7 | 6.0 ± 2.5 | hulled |
| LIT11 237 | pumpkin | yellow, orange | 3.6 ± 1.4 | 10.0 ± 3.3 | hulled |
| LIT11 251 | pumpkin | dark green, orange | 1.9 ± 1.2 | 5.1 ± 3.8 | hulled |
| LIT11 265 | scallop | green | 1.0 ± 0.2 | 7.4 ± 1.7 | hulled |
| LIT11 266 | zucchini | dark green | 2.0 ± 0.3 | 4.5 ± 1.8 | hulled |
| LIT11 272 | pumpkin | yellow, orange | 3.6 ± 1.5 | 6.5 ± 2.2 | hulled |
| ZAP12 001 | vegetable marrow, | dark green | 2.4 ± 0.5 | 9.7 ± 3.7 | hulled, hull-less |
| ZAP12 166 | zucchini, vegetable marrow | orange | 2.4 ± 0.4 | 11.1 ± 3.6 | hulled |
| ZAP12 288 | zucchini | orange, green | 1.8 ± 0.2 | 7.4 ± 1.4 | hulled |
| ZAP12 378 | vegetable marrow | cream, green | 2.7 ± 0.6 | 9.2 ± 2.7 | hulled, hull-less |
| 'Danka Polka' | pumpkin | cream | 2.8 ± 0.1 | 11.6 ± 0.9 | hulled |
| Overall average | - | - | 2.7 ± 1.2 | 9.0 ± 4.5 | - |

Explanation: Evaluation was performed at 'Wolica' Experimental Station, Warsaw, Poland. Polish pumpkin variety 'Danka Polka' was used as a reference variety. Varietal type was described based on Paris [23]. For fruit weight and yield values, standard deviations are given.

**Table 4.** Plant sex and mature fruit characteristics of cucumber accessions (*Cucumis sativus* L.).

| Accession Number | Sex Expression | Fruit Skin Colour | Fruit Spine Colour | Fruit Length (cm) | Fruit Diameter (cm) |
|---|---|---|---|---|---|
| LIT11 009 | monoecious | cream, brown | white, brown or black | 18.5 ± 1.1 | 7.3 ± 0.8 |
| LIT11 010 | monoecious | cream | white | 25.6 ± 1.3 | 8.6 ± 0.7 |
| LIT11 033 | monoecious | cream, brown | white, brown or black | 17.8 ± 0.7 | 7.7 ± 0.4 |
| LIT11 034 | gynoecious | cream | white | 20.1 ± 1.1 | 8.1 ± 0.4 |
| LIT11 035 | monoecious, gynoecious | cream, brown | white, brown or black | 20.5 ± 0.6 | 6.9 ± 0.6 |
| LIT11 065 | monoecious | cream | white | 18.0 ± 1.3 | 7.9 ± 0.9 |
| LIT11 079 | monoecious | cream, brown | white, brown or black | 18.7 ± 0.5 | 7.9 ± 0.6 |
| LIT11 128 | monoecious, gynoecious | cream, brown | white, brown or black | 20.7 ± 1.0 | 7.6 ± 0.6 |
| LIT11 147 | monoecious, gynoecious | cream, brown | white, brown or black | 18.0 ± 1.1 | 7.1 ± 0.4 |
| LIT11 161 | monoecious, gynoecious | cream, brown | white, brown or black | 18.5 ± 0.9 | 7.6 ± 0.7 |
| LIT11 189 | monoecious | brown | brown or black | 17.8 ± 0.8 | 8.3 ± 0.9 |
| LIT11 191 | monoecious | cream | white | 18.1 ± 1.0 | 8.1 ± 0.7 |
| LIT11 207 | monoecious | cream | white | 17.7 ± 0.8 | 7.8 ± 0.8 |
| LIT11 236 | monoecious | brown | brown or black | 16.3 ± 0.8 | 7.0 ± 0.5 |
| LIT11 252 | monoecious | cream, brown | white, brown or black | 18.1 ± 1.1 | 7.4 ± 0.9 |
| LIT11 257 | monoecious | cream | white | 24.7 ± 1.1 | 6.4 ± 0.4 |
| ZAP12 002 | monoecious | cream, brown | white, brown or black | 18.0 ± 0.9 | 7.2 ± 2.3 |
| ZAP12 003 | monoecious | brown | brown or black | 18.3 ± 0.6 | 7.9 ± 0.7 |
| ZAP12 051 | monoecious | brown | brown or black | 18.0 ± 0.7 | 7.9 ± 0.6 |
| ZAP12 057 | monoecious | cream | white | 18.7 ± 1.2 | 7.8 ± 0.6 |
| ZAP12 132 | monoecious | brown | brown or black | 18.7 ± 1.2 | 7.8 ± 0.5 |
| ZAP12 154 | monoecious | brown | brown or black | 18.7 ± 1.2 | 7.8 ± 0.5 |
| ZAP12 170 | monoecious | brown | brown or black | 18.5 ± 1.1 | 7.6 ± 0.7 |
| ZAP12 242 | monoecious | cream | white | 18.5 ± 1.1 | 7.6 ± 0.6 |
| ZAP12 248 | monoecious | cream | white | 18.5 ± 1.1 | 7.6 ± 0.4 |
| ZAP12 264 | monoecious | cream | white | 17.1 ± 3.5 | 6.9 ± 0.2 |
| ZAP12 299 | monoecious | cream | white | 19.0 ± 1.0 | 6.9 ± 0.6 |
| ZAP12 322 | monoecious | brown | brown or black | 17.8 ± 1.8 | 7.7 ± 0.9 |
| ZAP12 324 | monoecious | cream, brown | white, brown or black | 17.0 ± 0.9 | 7.5 ± 0.7 |
| ZAP12 331 | monoecious | cream | white | 17.1 ± 0.7 | 7.2 ± 0.7 |
| 'Trakai' | monoecious | brown | brown or black | 17.3 ± 1.0 | 8.3 ± 0.3 |
| Overall average | - | - | - | 18.7 ± 2.2 | 7.6 ± 0.8 |

Explanation: Evaluation was performed at 'Wolica' Experimental Station (2014). For fruit length and width, average with standard deviation (SD) is provided. 'Trakai' was used as a reference variety.

Within the accessions of *C. maxima*, a diversity of fruit skin color, weight, yield per plant, and seed color was observed. *C. maxima* fruits had various skin colors (from green, through grey, orange, to various shades of pink and red) and they often had two or more colors. In the Dzūkija Region, it was observed that whole mature fruits of winter squash

were soured together with cabbage. A single accession of *C. lanatus* was collected in the Mažoji Lietuva Region. The plants were grown undercover in a small foil tunnel. The fruit size was about 2 kg, and it had green skin and pink flesh. According to the grower, the seeds were brought from Georgia.

### 4.2.2. Solanaceous Vegetables

The third numerous group of annual vegetable plants collected (117) was represented by two solanaceous species, *Solanum lycopersicum* L. (98) and pepper *Capsicum annuum* L. (19). Collected accessions of tomatoes come mainly from the north-eastern (Aukstaitija, 39 accessions) and the north-western (Žemaitija, 39 accessions) parts of Lithuania. They were cultivated there in home gardens undercover, in small greenhouses or foil tunnels.

The collected forms of tomato varied in many important performance features: growth type (with a reduced number of side branches, high-growing), the earliness of fruit ripening (from very early to very late), fruit size (from cherry type to very large), fruit shape (round, flattened and oblong), fruit color (yellow, orange, red, raspberry and so-called black) and taste. They often have local names, e.g., 'Grandma's tomato', 'fingers' or 'morning rose'.

Nineteen (19) accessions of pepper (*Capsicum annuum* L.) were also collected. The majority of pepper accessions were collected in the Aukstaitija (11), Suvalkija (3), Mažoji Lietuva (3), and Žemaitija (2) regions. No cultivation of peppers was found in the Dzūkija Region. The fruits of the forms collected in other regions were red or yellow and block-shaped, most often sweet. A few accessions had short narrow pods and a pungent taste. They were grown in home gardens undercover, in small tunnels or greenhouses, together with tomatoes. Open field pepper cultivation was not observed in any of the regions.

### 4.2.3. Biennial and Perennial Vegetables

During the expeditions, 160 accessions of local populations or varieties belonging to biennial vegetable plants were collected; 12 accessions of perennial vegetables were also collected (Table 1). The group of biennial and perennial vegetables was dominated by populations belonging to the Alliaceae family, with species such as: *Allium sativum* L., *Allium cepa* L. var. *cepa* Herm., *Allium cepa* L. var. *aggregatum* G. Don., *Allium × proliferum* (Moench) Schard., *Allium ampeloprasum* ssp. *porrum* J. Goy, *Allium fistulosum* L., and *Allium schoenoprasum* L. (Table 1).

The majority (81) of accessions collected included garlic (*Allium sativum* L.). Great biological diversification was observed in the obtained material of local populations and varieties of garlic. The collected garlic bulbs consisted of 4–12 single cloves arranged in a regular ring or an irregular spiral. The bulbs were oblong, spherical or flattened, white, grey, pink, or purple, sometimes with anthocyanin streaks. They also varied in size: from 10 to 60 g. The majority of local populations of garlic were collected in the Žemaitija (34 accessions) and Aukstaitija (21 accessions) Regions, where garlic was common in home gardens.

Other species whose accessions were also collected in great numbers (51) was *Allium cepa* L. var. *cepa* Herm., i.e., common onion. The majority of onions were collected from the local populations grown in home gardens in the Žemaitija (26) and Dzūkija (14) Regions. The collected bulbs were of various sizes (small, medium, large, or very large), and of various shapes (flat, round, or oval). The color of the outer skin was very diversified; it was white, brown, red, straw-yellow, straw-pink, pink and brown, light yellow, brown and yellow.

The botanical variety of common onion (*Allium cepa* L. var. *aggregatum* G. Don.) occurred most frequently in the Žemaitija Region (6 accessions), where it is known as the shallot. Collected accessions of shallot varied in bulb size (small and medium), outer skin coloring (brown, yellowish, dark brown), and the number of cloves comprising the bulb (from 4 to 11).

Four accessions of local varieties of the top-setting onion, *Allium × proliferum* (Moench) Schard., were also collected in the Žemaitija Region. Bulbils forming at the top of the flowering scape (stem) were collected. Top-setting onion was grown in home gardens as a perennial plant (4–6 years in one place), which started the growing season very early.

According to the farmers, it is the first bulb plant whose green chives were very tasty and fit for consumption early in spring. The bulbils which formed gradually during the plant growth were rarely used for consumption—they were used as a multiplication material.

Single accessions of three perennial species were collected and they belonged to the Alliaceae family: *Allium schenoprasum* L. (chives) and *Allium fistulosum* L. (brunching onion) as well as one species of biennial *Allium ampeloprasum* ssp. *porrum* J. Goy (leek).

In the group of biennial vegetables, seeds of single local populations of the Apiaceae family were collected: carrots—*Daucus carota* L. (1), common parsnip—*Pastinaca sativa* L., (1) and common parsley—*Petroselinum sativum* L. ssp. *crispum* (4), from the Chenopodiaceae family: red beet—*Beta vulgaris* L. (9), from the Brassicaceae family: head cabbage—*Brassica oleracea* L. var. *capitata* L. f. *alba* (2) and cultivable swedes—*Brassica napus* L. var. *napobrassica* L. Peterm. (1) as well as from the Polygonaceae family—rhubarb—*Rheum rhaponticum* L. (1).

*4.3. Cereals, Forage, Industrial and Fibrous Crops as Well as Segetal Plants Included in the Crop Collected Material*

A total of 125 accessions of cereal plants were collected and some of them were blends. The blends were very well prepared by the local populations. The majority of them are cereals grown for feeding. More detailed results for common oat were obtained, for which evaluation was conducted in a 2-year field experiment (2012 and 2013) with the material collected in 2011 and 2012. The accessions collected in 2013 were not included in the results of this article because they come only as mixtures.

Oat *Avena sativa* L.

During the collecting missions organized between 2011 and 2013 in Lithuania, seeds of 33 local oat varieties were collected. The collected oat was cultivated mainly for fodder, some accessions came from cereal mixtures. Among the 9 accessions collected in 2011, four (LIT11 138, 139 144, 168) were grown for food production, the remaining ones for fodder. In 2012, 19 accessions were collected, two of them were used for sprouts (ZAP12 338 and 360), and the rest for fodder, including 6 accessions separated from mixtures with barley (ZAP12 006, 383, 389, 390, 289, 309). In 2013, only 5 accessions of local oats were collected, which came from fodder mixtures with barley, rye, wheat, and peas, and they were not included in the field experiments.

Accessions collected in 2011 and observed in 2012 presented a large variability with low individuals (95 cm) and high plants up to 132 cm (Figure 4A). Among the accessions of *A. sativa* collected in 2011 and observed in 2012, on 2.5 m$^2$ plots, the accession LIT11 139 was most distinct. Its panning phase was the earliest and at the same time, it was one of the fastest maturing while the individuals were characterized by high TGW (39 g) (Figure 5A). Lodging did not occur and no diseases (no powdery mildew) were observed (Figure 6A). The accession was characterized by the highest thousand grains weight (TGW 39 g), but present the lowest yield (0.52 kg/2.5 m$^2$), despite the lack of lodging and high resistance to disease, his harvest was only 30% of the yield of the reference variety 'Krezus'. The others observed accessions had also a high resistance to diseases: there was not observed powdery mildew infection, just medium crown rust infection in two accessions: LIT11 144 and LIT11 186 (Figure 6A).

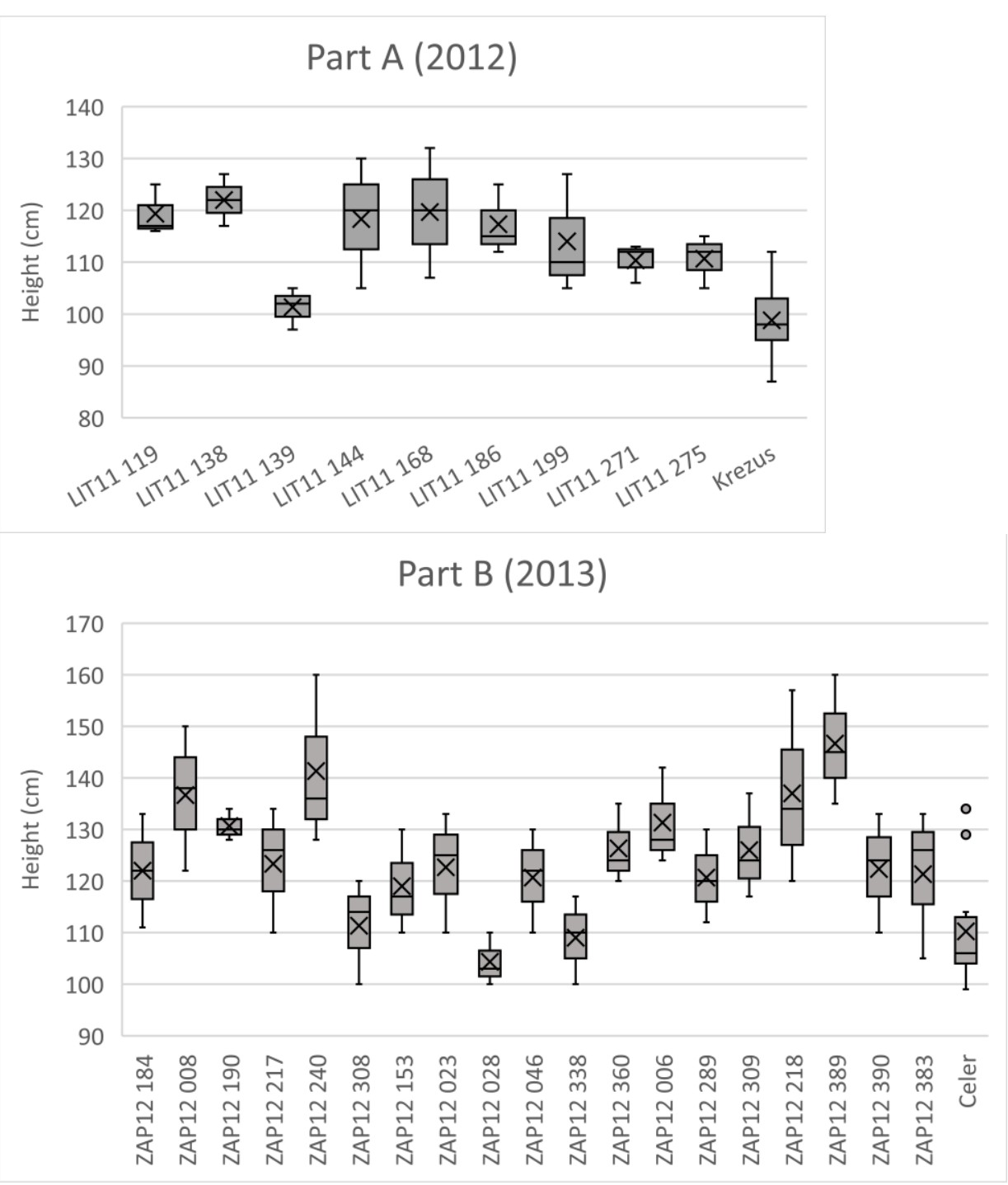

**Figure 4.** Variation in height of oat landraces gathered in Lithuania in 2012.

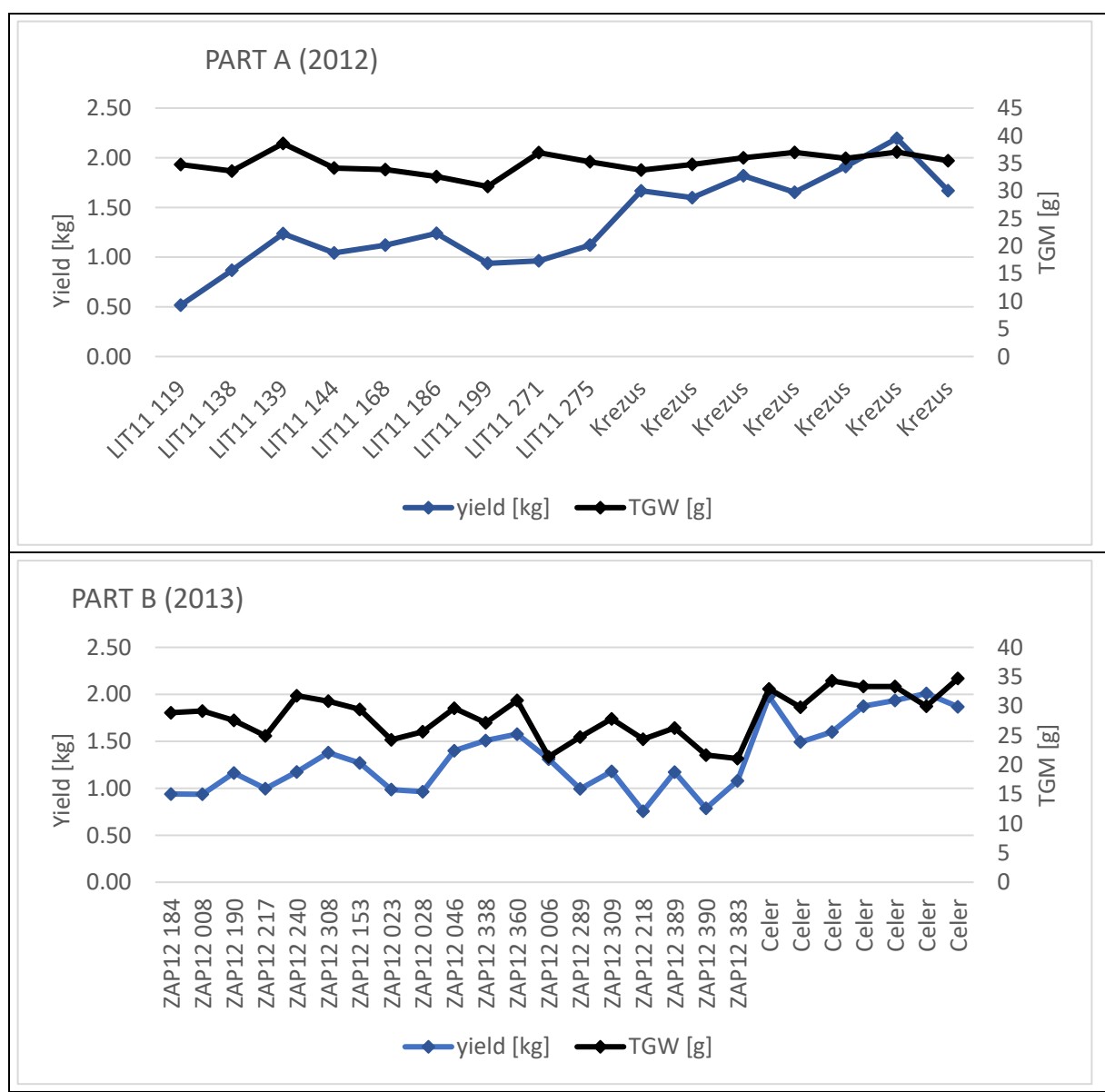

**Figure 5.** Yield parameters of *Avena sativa* L. accessions gathered in Lithuania and evaluated in 2012 and 2013 (modified from Kloc and Dostatny, 2020) [27].

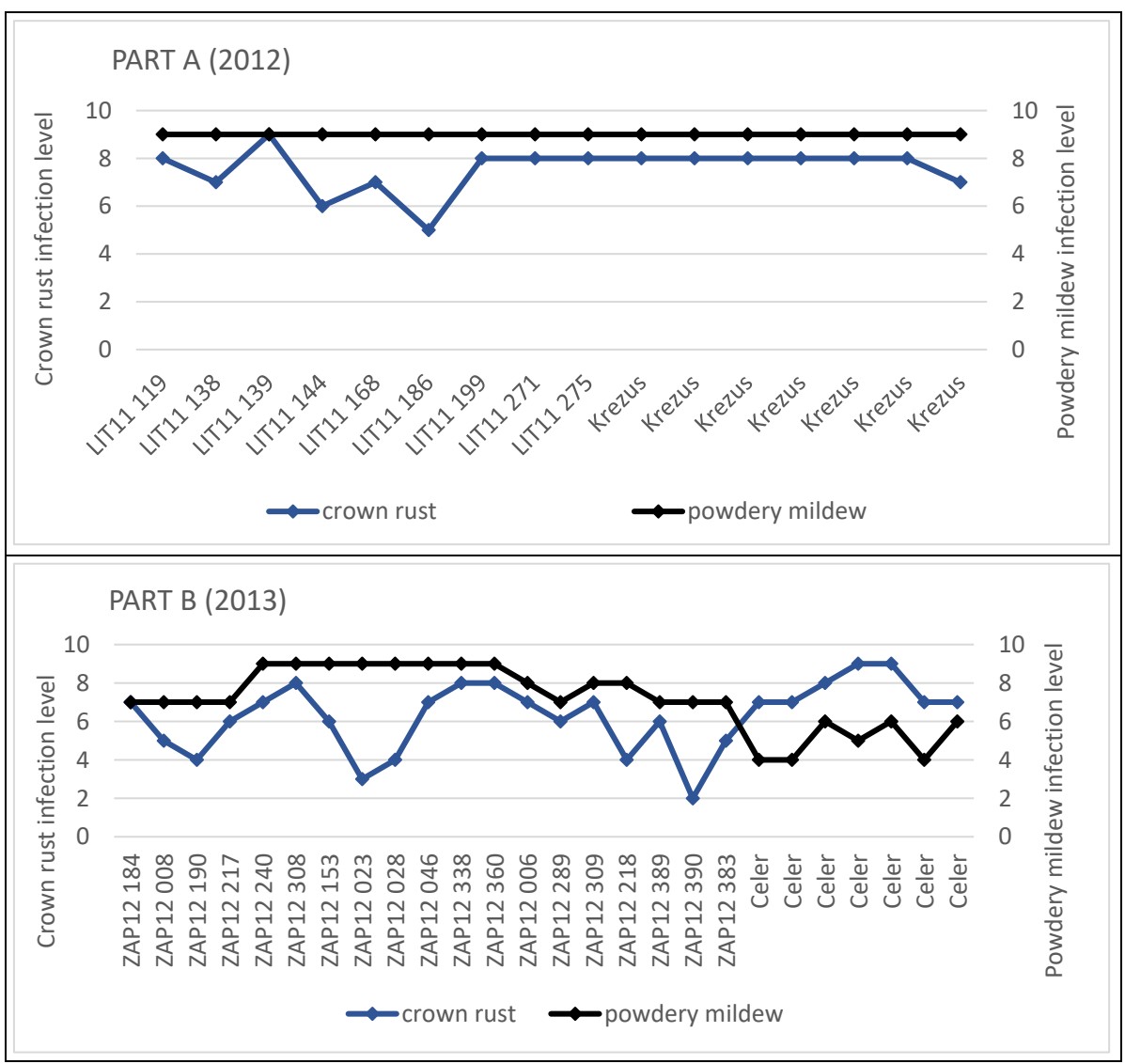

**Figure 6.** Resistance to powdery mildew and crown rust in cultivated *Avena sativa* L. gathered in Lithuania and evaluated in Poland in 2012 and 2013 (modified from Kloc and Dostatny, 2020) [27].

Accessions collected in 2012 and observed in 2013 presented a large variability with low individuals (100 cm) and high plants up to 160 cm (Figure 4B). For the 2012 population accessions collected in Žemaitija, the observed trend was not similar to the populations collected in other regions: the lower individuals did not always have high TGW and most of them had low yields, most likely because they were either blended (i.e., came from cereal mixtures) or grown for feed, as a forage plant. During observations conducted in 2013, a yield from 1 to 1.58 kg/2.5 m$^2$ was observed and the TGW ranging from 19 to 33 g (Figure 5B) was noted. Eight (8) out of 19 accessions showed no infection with powdery mildew, whereas crown rust was observed in most of the accessions (Figure 6B). While in reference varieties, powdery mildew infection was found in the range between 4 and 6, in the case of collected local varieties it was between 7 and 9.

The segetal plants constituted an additional group that was not included in the table, as they were separated from uncleaned cereal material and not collected as separate accessions. Thirty-eight (38) weed species were found. Among segetal plants in collected accession cereals, *Chenopodium album* L. was the species occurring most often. However, *Galium aparine* L., *Fallopia convolvulus* (L.) Á. Löve, *Artemisia vulgaris* L., *Galeopsis tetrahit* L., *Polygonum lapathifolium* L., *Stellaria media* (L.) Vill., *Viola arvensis* Murray also occurred frequently.

## 5. Discussion

Plant germplasm is collected in order to preserve it, as well as to conserve and expand the genetic base that can be used in breeding programs or can have potentially direct use [28]. Bioversity International supported a series of collecting expeditions worldwide, with the objective to systematically collect and conserve landraces cultivated by farmers and their crop wild relatives which were lost from fields and natural habitats [7]. Plant collecting activities date back to the beginning of agriculture, with the first steps of plant domestication [29].

Situated in the Neman Basin of the Baltic Sea, Lithuania is one of the Eastern European countries which shares its south-western border with Poland. The geographical situation of Lithuania and the relatively severe climate in this part of Europe results in low diversity of plant genetic resources; however, in Lithuania tradition of home gardens exists, and such gardens are recognized as hotspots of agrobiodiversity [30]. Recently, locally grown landraces and populations, including home gardens, have been substituted by modern varieties, thus collaborative expeditions to protect the diversity of Lithuanian local crop biodiversity were organized.

The interview was an important part of the expedition as, for example, one can learn about the ripening time of fruit trees' fruits, important performance parameters of the collected local varieties of vegetable and agricultural plant species, and can find out how individual plants grown by a farmer are used. It was confirmed in the publication of Genebank Standards [21].

Later maturing time and the selection of varieties occurring in Lithuania result from the climatic and soil conditions, while the economic and cultural factors contribute to the high occurrence of old fruit trees, whereas the reserve of species and varieties results from the local climate. The maturing time of apple tree varieties found in Lithuania is also different from the maturing time of the same varieties in the climatic conditions of Poland. Mentioned varieties of apple trees cited in this article are relatively tolerant to lower temperatures, which determines their scope in the environmental conditions of Lithuania.

Varieties of apple trees that were cataloged in Lithuania were bred in that country or came from neighboring countries having similar climates. The biggest source of the multiplication material included orchards remaining from old estates or left by known growers who lived in the area of Lithuania at the turn of the 19th and 20th centuries. An eminent biologist, breeder, and pomologist, Hrebnicki (1857–1941) was one of them, and he published a pomology, 'Atlas owoców' (Fruit atlas) in Polish and Russian in Saint Petersburg in 1906 [31]. Hrebnicki grew and described many varieties of apple trees, which are not only known and grown in Lithuania but also in other European countries. The researcher collected about 512 varieties of apple trees, 256 varieties of pear trees, and about 100 varieties of plum trees as well as many other species and varieties of fruit plants in the Rous Estate near Dukštas in the Aukstaitija Region. Trees gathered there came from Lithuania, Latvia, Estonia, Belarus, Russia, Ukraine, Poland (from the orchards of P. Hoser, E. Jankowski, and J. Ślaski), and from Germany and North America. Apart from known apple tree varieties occurring almost throughout Europe, there were some very rare ones known only in Lithuania, bred or described by Hrebnicki, which were listed in the results of this article.

Varieties obtained during the expeditions to Lithuania made it possible to learn about the abundance of fruit crops and their health. The results show that many varieties occur only in the area of Lithuania and they are the so-called local varieties which, according to the literature, were grown or found in this territory within specific places and can only be encountered there. 'Paniemuńskie Białe' is one of such varieties. According to Tuinyla et al. [2], it comes from a place called Panemunia near Kaunas and was first described by Hrebnicki. It occurs not only near Kaunas but also in the rest of the country. Another very old variety found in Lithuania is 'Cukrówka Litewska' (Figure 2C,D), most often occurring in the Alytaus Region, which is confirmed in the description by Smardzewski [32]. The origin of this variety is unknown; it is very vital, which is confirmed by the occurrence of

very old specimens whose age was determined to be about 100 years during the expedition. 'Birutės pepinas' is another variety associated with this area of Europe. Tuinyla et al. [2] wrote in their description that it comes from the vicinity of Kaunas, but it can also be found in other regions of the country; however, it is not as popular as 'Pepinka Litewska' from which it originates. 'Pepina Jana' is a variety that was found and described by Hrebnicki [31]. It occurs mainly in the Šiaulių Region; however, to a smaller extent, it can also be found in other regions of Lithuania. It does not typically occur outside Lithuania. 'Śmietankowe' is a very interesting variety with delicious fruits. According to Smardzewski [33], it comes from Žemaitija, but very frequently it can be found in the north of Lithuania, in the Šiaulių Region. Another variety that is very characteristic for the area of Lithuania is 'Malinówka Bierżenicka'. According to Hrebnicki [31], at the beginning of the 20th century, it occurred in large numbers in the Ignalinos Region, but presently it is very rare there; during the expedition, it was found only once in the Vilniaus Region.

There is a group of fruit tree varieties, which quite often occur in old orchards in Poland and other European countries. They come mostly from countries bordering the Baltic Sea and Russia, but also from France, the Netherlands, and Germany. In these varieties, a significant difference can be observed between the fruit maturing time in Lithuania and the time given in the literature. 'Ananas Berżenicki', 'Antonówka Zwykła', 'Pepinka Litewska', 'Grafsztynek Inflancki', 'Kronselska', 'Oliwka Żółta' ('Papierówka') and 'Suislepskie' are examples of such varieties. The difference in maturing time is from three to four weeks, depending on the variety. According to Smardzewski [32] and Rejman [34], 'Grafsztynek Inflancki' matures at the end of August and at the beginning of September, while in Lithuania it happened at the end of September and at the beginning of October. Another variety that is very high in the popularity structure of the Lithuanian varieties is 'Pepika Litewska.' In Lithuania, the fruits mature at the beginning of October, whereas, the maturing time in Poland is in mid-September [34]. According to Rejman [34], 'Oliwka Żółta' and 'Suislepskie' are ready for consumption in Poland already at the end of July and at the beginning of August, while in Lithuania—at the end of August and at the beginning of September. 'Kronselska' deserves special attention. This variety is of French origin and according to Smardzewski [32] it was grown in Cronsels near Troyes. Despite the fact that it comes from areas with a warmer climate, it does very well in Lithuania and is often found in old orchards. According to Rejman [34], in Poland, the 'Kronselska' variety matures at the beginning of September, whereas in Lithuania at the end of September and at the beginning of October.

The area of Lithuania constitutes a northern border of cultivation of several vegetable crops including cucurbits, beans, and tomatoes. Collection of vegetable accessions in Lithuania fulfilled the gaps in germplasm collections representing Central and Eastern Europe. Regions that are most affluent in various vegetable plant genotypes included Žemaitija, Aukstaitija, and Dzūkija.

Dzūkija is the region in which the cultivation of pumpkin, squashes, and cucumbers in home gardens was observed most often. The highest number of cucurbits—*C. pepo* (22), *C. maxima* (21), and *C. sativus* (17)—was collected there which constituted 36%, 40%, and 54% of all collected accessions for these species during the three expeditions to Lithuania (Table 1), respectively. This is probably due to the climatic conditions favorable for the cultivation of stenothermal plants in this region. In Lithuania, different varietal types of *C. pepo* pumpkins and squashes are grown. The majority of *C. pepo* accessions were collected in the Dzūkija (22), Žemaitija (15), and Aukstaitija (14). High diversification of performance parameters was observed in the individual accessions of *C. pepo*. This is due to the fact that *C. pepo* is an entomophilous species that crosses easily within botanical varieties. In Lithuania and Poland, certain types of *C. pepo* pumpkins are called and recognized as 'Artroka'. The authorship of the 'Artroka' variety is attributed to a farmer named Hejbowicz, who—at the beginning of the 20th century—obtained from Argentinian Indians pumpkin seeds with strong medicinal properties. However, the fruits of this pumpkin did not mature in the Lithuanian climate; therefore, he crossed it with a cold-tolerant variety

from the area of Trakai. Hejbowicz called the new variety 'Artroka' (a combination of words Argentina and Trakai) and emphasized that it inherited all the medicinal properties from the Argentinian Indian variety. Landrace 'Artroka' was characterized by dark green fruit skin color with yellow spots, smooth surface or with little bumps and the flesh was yellow and orange with a lot of hulled seeds. It seems that the 'Artroka' landrace does not exist anymore because it was not maintained; however, this type of fruit can be observed in some local pumpkin populations grown by farmers in Lithuania and Poland [35]. Collected *C. pepo* accessions could be carefully studied and possibly useful to restore this landrace. There are examples of successful restoration of local *Cucurbita* landraces for example 'Berrettina di Lungavilla' in the Po valley in northern Italy [36].

Most of the cucumber accessions collected during the expeditions showed characteristics typical for the 'Trakai' variety. Plants were monoecious with medium-sized (about 18 cm long) dark-thorned fruits. Only two accessions were characterized by longer fruits (about 25 cm), one was gynoecius and four were segregated for sex type (Table 4). The 'Trakai' variety, famous in Lithuania, comes from the vicinity of the Trakai town near Vilnius and is closely related to Karaimes. The 'Trakai' cucumber is considered to be introduced in the 15th century when the Karaimes came to Lithuania from the Crimea and specialized in growing 'oriental' vegetables including cucumber [37,38]. In 1844, Strumiłło described 'Trakai' cucumbers as equally fruitful and juicy as the cucumbers of Russian variety 'Muromskij' but bigger, longer, and better for traditional pickling in brine. He also wrote that Karaimes and Tartars living in the surroundings of Vilnius grew them for sale in Vilnius [39]. As mentioned by Krywko in 1926 'Trakai' cucumber (locally called also Karaimes' cucumber) was listed in the Welter's seed catalog [38]. The variety was also well-known and grown in central Poland from the 1930s to the 1970s until the time when the first high-yield hybrid varieties characterized by non-yellowing fruits with white thorns were introduced. Recovery of this landrace could be considered; however, it is monoecious and susceptible to major diseases (unpublished data), thus it will require extensive disease management and it will be low-yielding as compared to modern hybrids.

Tomato and *Capsicum* seeds were collected from small plastic or glass greenhouses located in the backyards. In the Lithuanian climate, tomatoes and pepper are cultivated only in conditions that are rather unfavorable (limited chemical protection, changes in light intensity and temperature, accumulation of pathogens and fertilizers in the soil). Local growers collect, for the next season, seeds from plants that are healthy and fruits whose taste is preferred locally. Thus, collected accessions could be an interesting source of resistance/tolerance to both biotic and abiotic stresses. Such germplasm could be useful to study multiple stress resilience mechanism(s) to develop solanaceous varieties for organic or low-input agriculture. Combined multiple stress tolerance is of great interest [40,41] and there are efforts to achieve resilient tomatoes that keep unique flavor [42,43]. Therefore, valorization of collected accessions with a focus on stress tolerance and description of locally adapted flavor would be of great interest.

Collected accessions of garlic and onions could be characterized by early resistance/tolerance to stresses and long storage overall high adaptability for Eastern European climatic conditions. Growers used to collect the bulbs and dry them in simple conditions. Due to extensive global garlic and *Alliums* exchange, more serious viral infections of those vegetables have been observed recently [44–46]. Thus, it would be interesting to analyze bulb storage parameters of the accessions and virus resistance of the accessions.

Whole bean plants with pods drying on the fences or under the eaves of outbuildings were a frequent view in the Aukstaitija and Žemaitija Regions. A similar sight can still be encountered in some regions of eastern and southern Poland [47]. Field beans are grown in this region in a similar way as beans, to obtain dry seeds, and they are harvested in the stage of green immature seeds for direct consumption only in a small percentage, unlike in Poland or other regions in Europe, where field beans are mainly used in the form of immature seeds. It is observed that field beans are used in a similar way as in Lithuania in Central America, Egypt, Ethiopia, and China where dry field bean seeds constitute the



basic source of protein in the nutrition of people and form the basis of their everyday diet. High protein content in seeds is one of the most important indicators of the nutritional value of beans. Beans are good to excellent sources of protein, fibers, carbohydrates, and micronutrients, which often are lacking in diets [48,49]. Human plant protein intake is on the rise in many EU regions and the market for meat and dairy alternatives is undergoing annual growth rates of 14% and 11% respectively. The characterization and maintenance of food legume genetic resources and their exploitation in pre-breeding form the core development of both more sustainable agriculture and healthier food products [50]. The collected material can be a rich source for further exploitation.

It is puzzling that not a single accession of common pea (*Pisum sativum* L.), either field (dry) pea or garden (vegetable) pea, was found (only as a field pea, but in the composition of the feed mixture). Also, no cultivation of tomato was observed in open soil, which is understandable, as the climatic conditions in northern Lithuania do not allow the growth of such a stenothermal plant as a tomato.

According to Arlauskienė et al. [51], the highest concentration of cereals (up to 80%) was found in the crop production farms of Central Lithuania (31–100 ha), but oats have been grown on the least productive soils (in Western and Eastern Lithuania). In recent years, organic farming oats have been grown all over Lithuania, while in western Lithuania, because of adverse wintering or poor autumn conditions, farmers tend to choose spring cereals instead of winter ones. During the collection missions, more cereals' accessions were collected in Eastern Lithuania (Žemaitija Region), but also in the southern and central part of Lithuania (Dzūkija and Suvalkija Regions) due to the fact that the goal of these expeditions was to collect local varieties, instead of new varieties which are grown mainly in central Lithuania.

Cereals and legume mixtures for forage production are very popular in Lithuania. The share of legumes in the mixture has a positive effect on the value of the site for successive crops, and pea seeds, rich in protein, increase their fodder efficiency. In addition, these mixtures are less susceptible to weed infestation and infestation by diseases and pests, which makes them less demanding in relation to the expenditure on chemical plant protection products [52]. The segetal plant species found in the collected accession cereals are common in cereal intercrops in Lithuania [51]. No rare species of weed were found.

The observed oat landraces plants from both 2012 and 2013 experiments conducted in Radzików were characterized by a large diversity of phenotypic traits. In the plots in 2012, low individuals were characterized by high TGW, whereas high individuals had low TGW. In the experiment conducted in 2013, the landraces of oat plants were very variable. The results of the observations do not allow to state the existence of any stable correlation between characters of the observed collected accessions—it can be said that each of them is individual. According to Kordulasińska and Bulińska-Radomska [53] and Pszczółkowski and Sawicka [54], a large variety of traits opens the way for the selection of beneficial traits for breeding. Varieties with the best morphological or agricultural traits (collected in 2011) could be used in the breeding process to create genotypes with the desired traits [55].

Oat landraces have considerable potential for improving disease and abiotic stress tolerance, and additionally, there are no obstacles to transfer the favorable characteristics from the landraces to new varieties of crops. Oat landraces provide a wide spectrum of starting materials for plant growth, as they make it possible to choose the characteristics which are best suited to local needs. They are adapted to grow in the local environment and local climate and immune to local diseases. Iannucci et al. [56] also emphasize that the lack of strong links between the observed traits allows obtaining 'useful combinations' for breeding. Besides that, a high level of variability in the analyzed indigenous oat landraces increases their ability to compete with weeds. Therefore, it has to be concluded that oat landraces should also be grown and conserved on farms. This, however, requires both scientific expertise and traditional knowledge of farmers who should grow and manage oat landraces in the place where they originated.

Collection missions are important sources of germplasm that can be used, among others, in organic farming production and other sustainable agriculture programs, which is necessary in the face of global climate changes and the need to ensure local and global food security [28]. Genetic diversity is the key to maintain and improve agriculture.

## 6. Conclusions

The main purpose of collecting plant genetic material is to create collections that represent the widest possible genetic diversity of the gene pool of a given population. Field expeditions that took place in the area of Lithuania between 2011 and 2013 allowed us to get acquainted with the abundance of this country in the remnants of old orchards, where interesting fruit tree varieties, mainly apple trees grown at the turn of the 19th and 20th centuries, can be found even today. During the expeditions, attention was paid to the diversification of species and the age of trees. The geographical situation of Lithuania and the relatively severe climate in this part of Europe definitely influence a low diversity of fruit tree varieties and a limited number of such varieties. Despite the climatic conditions, which are not favorable for growing fruit crops, a relatively big pool of very old fruit trees was observed. There is no program for the preservation of old varieties of fruit trees in Lithuania; therefore, there is also no awareness allowing to save them. For many apple tree varieties, it was the last moment to save them by grafting. Poor health of trees was often an obstacle in collecting material for multiplication. Field expeditions to Lithuania proved to be appropriate and of great value, as they allowed to localize precious genetic resources, such as many old apple tree varieties becoming extinct (the collected material will be used for new planting and, thus, saving the diversity found there).

A part of the collected material can serve as a basic material for plant breeding, in search of a better adaptation of these species to different soil and climatic conditions.

Germplasm collecting in Lithuania resulted in the preservation of many genetic plant resources, representing a high genetic variability in the country, which are now safely conserved and maintained for the benefit of present and future generations.

**Author Contributions:** Conceptualization, D.F.D.; methodology, D.F.D., A.K., R.R.; formal analysis, D.F.D., G.B., A.K., K.K.; investigation, D.F.D., A.K.; data curation, D.F.D., A.K., G.B., B.G.; writing—original draft preparation, D.F.D., A.K., R.R.; writing—review and editing, D.F.D., G.B.; visualization, D.F.D., G.B., K.K.; supervision, D.F.D.; funding acquisition, D.F.D., G.B. All authors have read and agreed to the published version of the manuscript.

**Funding:** This work was supported by the Multi-annual program: "Establishment of a scientific basis for biological progress and preservation of plant genetic resources as a source of innovation in order to support sustainable agriculture and food security of the country" coordinated by Plant Breeding and Acclimatization Institute—National Research Institute (PBAI-NRI) and financed by the Ministry of Agriculture and Rural Development of Poland.

**Institutional Review Board Statement:** Not applicable.

**Informed Consent Statement:** Not applicable.

**Data Availability Statement:** Not applicable.

**Acknowledgments:** We thank Wiesław Podyma, the head of the National Research Institute (PBAI-NRI), who provided a factual review of the manuscript, and Valdemaras Degulis who performed the duties of a guide, translator, and assistant during all three expeditions.

**Conflicts of Interest:** The authors state no conflict of interest.

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
