# Peer review of "The Evaluation and Conservation of Plant Genetic Resources Collected in Lithuania"

_agronomy, doi:10.3390/agronomy11081586_

Round 1
Reviewer 1 Report
The paper e is devoted to expeditionary surveys of Lithuania, collection of plant genetic resources, their study and preservation.
It is quite informative, describes in detail the survey areas, the collected material. However, it gives the impression of being overly verbose. There are many well-known concepts that could be avoided. I suggest shortening the text by at least 10%.
Errors noticed:
- Lines 246-276 ??? What is this: an excerpt from the rules for authors?
- In the text, the Latin names of plants are not italicized (for ex., lines 369-370; 732, etс.), only in the table 1.
- Line 732: “not a single sample of common pea (Pisum sativum L.), either scaly pea or sugar pea”. There are no such definitions of pea. They distinguish: dry (field) pea, forage (fodder) and vegetable (garden) pea. Either there is no field wheat (line 496)ю
- Instead the word “sample” it is more usual to use the term “accession” for the collections of germplasm.
What descriptors the authors used when they evaluated the collected material?
Author Response
As suggested by the reviewer, the entire article has been shortened and the well-known concepts were deleted.
- Lines 246-276: Chunked in text. It has been deleted.
- All Latin names of plants are in italic now.
- Line 732: The mistake in the naming of pea and wheat has been corrected.
- Done
- Answer the question about descriptors: Most of the descriptors were used according to the descriptors used by the ECPGR working groups. If other descriptors were used, they were given in the methodology. For oats, the resistance scale was used instead of the susceptibility (where 9 is a healthy non-lodging plant, and 1 refers to a less resistant, lodging plant) according to the guidelines used in Poland and other eastern European countries.
Reviewer 2 Report
The Authors present the work done well and it appears quite interesting for future applications. However, a molecular identification and characterization of all the material is currently useful and indispensable.
It is recommended that you move the study area (paragraph 2) to the materials and methods paragraph as the first section.
There is a large portion of lines 247 through line 274 of the authors instructions that must be deleted.
Author Response
We thank the reviewer for having carefully read the manuscript and for all his suggestions. Below you can find the answers for your questions:
The collected material must firstly be regenerated and evaluated in the fields and then a molecular identification and characterization of the material need to be conducted. Some of the accessions are currently in the process of molecular identification. Unfortunately, such research requires a larger budget, which we do not always have, and this research stretches out a little over time. However, as soon as they are ready, they will be available.
The fragment in the second paragraph in the "Study area" that referred to the "Methodology" has been deleted.
The fragment with authors instructions was a mistake and has been deleted.